# Atmospheric teleconnections between the Arctic and the Baltic Sea region as simulated by CESM1-LE

Erko Jakobson[1] and Liisi Jakobson[1]

[1]Tartu Observatory of University of Tartu, Tartu, Estonia

*Correspondence to*: Erko Jakobson (erko.jakobson@ut.ee)

**Abstract.** This paper examines teleconnections between the Arctic and the Baltic Sea region and is based on two cases of CESM-LE climate model simulations': the stationary case with pre-industrial radiative forcing and the climate change case with RCP8.5 radiative forcing.

Stationary control simulation 1800-year long time-series were used for stationary teleconnection and 40-member ensemble
from the period 1920–2100 for teleconnections during ongoing climate change. We analysed seasonal temperature at a 2-meter level, sea-level pressure, sea ice concentration, precipitations, geopotential height and 10-meter level wind speed. The Arctic was divided into seven areas.

The Baltic Sea region climate has strong teleconnections with the Arctic climate; the strongest connections are with Svalbard and Greenland region. There is high seasonality in the teleconnections, with the strongest correlations in winter and the lowest
correlations in summer, when the local meteorological factors are stronger. NAO and AO climate indices can explain most teleconnections in winter and spring. During ongoing climate change, the teleconnection patterns did not show remarkable changes by the end of the 21st century. Minor pattern changes are between the Baltic Sea region temperature and the sea ice concentration.

We calculated the correlation between the parameter and its Ridge regression estimation to estimate different Arctic regions'
collective statistical connections with the Baltic Sea region. Seasonal coefficient of determination, $R^2$, were highest for winter: for T2m $R^2 = 0.64$, for SLP $R^2 = 0.44$ and for PREC $R^2 = 0.35$. When doing the same for the seasons' previous month values in the Arctic, the relations are considerably weaker, with the highest $R^2 = 0.09$ for temperature in the spring. Hence, Arctic climate data forecasting capacity for the Baltic Sea region is weak.

Although there are statistically significant teleconnections between the Arctic and Baltic Sea region, the Arctic impacts are
regional and mostly connected with climate indexes. There are no simple cause-and-effect pathways. By the end of the 21st century, the Arctic ice concentration has significantly decreased. Still, the general teleconnection patterns between the Arctic and the Baltic Sea region will not change considerably by the end of the 21st century.

# 1 Introduction

The Arctic region is warming at least twice (IPCC, 2021; Nakamura and Sato, 2022; Overland et al., 2018; Meleshko et al., 2020), some authors showed that nearly four times (Rantanen et al., 2022) as fast as the whole planet; also, the Baltic Sea region is warming faster than the global average (BACC, 2015). The question of the faster warming in the Arctic region affecting mid-latitudes has been under debate for a long time. We want to address the Baltic Sea region and find out if it is affected by the changing Arctic and if this information can be used in long-term weather forecasts.

The faster warming in the Arctic compared to the global mean, a phenomenon known as Arctic amplification (AA), is a result from interacting processes: sea ice loss (Simmonds and Li, 2021) and surface albedo feedback (Laîné et al., 2016; Yoshimori et al., 2014; Serreze et al., 2009; Screen and Simmonds 2010), changes in longwave (Lee et al., 2017) and temperature feedbacks (Dai and Jenkins, 2023; Jenkins and Dai, 2021; Duan et al., 2019; Pithan and Mauritzen, 2014; Lu and Cai, 2009), cloud changes (Taylor et al., 2022; Boeke and Taylor, 2018; Taylor et al., 2015; Taylor et al., 2013; Francis and Hunter, 2006; Vavrus, 2004), intraseasonal cycling of heat (Clark et al., 2021; Bintanja and Krikken, 2016; Bintanja and Linden, 2013), and poleward energy transport (Sang et al., 2022; Spielhagen et al., 2011). However, the relative weight of these different factors is still under debate (Taylor et al., 2022; Dai et al., 2019).

The assessment of the potential for AA to influence broader hemispheric weather (referred to as teleconnections) is complex and controversial (Dai and Song, 2020; Francis and Vavrus, 2015; Barnes and Screen, 2015; Sun et al., 2016) and many details of teleconnection mechanisms remain elusive (Sun et al., 2018).

For a long time, it has been recognised that ice conditions in the Greenland region might be connected to several variables in Europe (Hildebrandsson, 1914; Wiese, 1924; Schell, 1956; etc.). Zhuo et al. (2023) found that the Greenland region is the most influential of all Arctic regions on teleconnections to the Baltic region. Deng et al. (2018) showed that the Greenland region was the most important of Arctic regions considering heat waves in Eastern Europe.

AA is expected to be related to further changes that affect mid-latitudes and the rest of the world (Jung et al., 2015; Vihma et al., 2019). According to Overland et al. (2015), potential Arctic teleconnections with Europe are less clear than with North America and Asia. The teleconnections between the Arctic and mid-latitudes depend strongly on the season and geographical region (Zhuo et al., 2023; Coumou et al., 2018; Jakobson et al., 2017). Furthermore, it has been recognised that extratropical impacts depend highly on the regional structure of the anomalous Arctic climate state (Kug et al., 2015). It appears that Arctic impacts will be regional and intermittent, clouding the identification of cause-and-effect and raising the issue of how to effectively communicate potential Arctic impacts (Overland et al., 2021; Rudeva and Simmonds, 2021; Luo et al., 2019; Cohen et al., 2018).

The Baltic Sea region is very sensitive to climate change; it is a region with spatially varying climate and diverse ecosystems (Christensen et al., 2022). Climate change may bring profound ecological changes in the region (Halkka, 2022). During the last half-century, the duration of seasonal snow cover and snow depth have decreased (Viru and Jaagus, 2020); during the last decades, there has been a major increase in both extreme mild ice winters and severe ice winters, minor increase in the intense

precipitations, heat waves and cold spells (Rutgersson et al., 2022). Because of the closeness to the Arctic, the Baltic Sea region receives influences from the Arctic either remotely (teleconnections) or directly. The weather in the region depends highly on the position of the polar front: it can be located northward as well as southward of the area (Jakobson et al., 2017). Furthermore, some direct impacts are influenced by remote processes in the Arctic. For example, possibly the Barents and Kara Seas warming associated with the sea ice loss affects the Ural blocking (Peings et al., 2023; Yao et al., 2017; Luo et al., 2017), which has been identified as precursors of sudden stratospheric warmings (Statnaia et al., 2020; Lee et al., 2019; Martius et al., 2009), and extreme temperature/precipitation anomalies over Europe (Yang et al., 2022; Peings, 2019; Cattiaux et al., 2010).

Widely used climate model simulations, such as those from the Coupled Model Intercomparison Project (CMIP6), combines internal and inter-model variability caused by differing physics, dynamical cores, and resolutions, making it almost impossible to assess the portion of uncertainty caused by internal variability alone (Kay et al., 2015). To enable quantification of internal variability in the midst of transient climate change, large ensembles with individual models have been performed. Comparison across ensemble members simulated with the same model and external forcing provides a measure of simulated internal variability. Here, we use the Community Earth System Model version 1 large ensemble (CESM-LE; Kay et al., 2015) to diagnose connections between the Arctic and Baltic regions. In the last few years, CESM-LE has aimed at better understanding internal variability (Rondeau-Genesse and Braun, 2019). The CESM-LE has been used in multiple studies of Arctic sea ice cover, performing well overall (Smith and Jahn, 2019; Labe et al., 2018; Jahn, 2018; Massonnet et al., 2018; Barnhart et al., 2016; Jahn et al., 2016; Swart et al., 2015). The CESM-LE also produces credible NAO interannual aspects, given the length of the observational record available for assessment (93 years) (Deser et al., 2017).

Our previous research studied Arctic-Baltic teleconnections and physical mechanisms behind Arctic-Baltic teleconnections (Jakobson et al., 2017), using ERA-Interim and NCEP-CFSR reanalyses. In this paper, we used the CESM-LE model time-series to verify previous results, examine climate change's influence on teleconnections and study different Arctic regions' collective forecasting capabilities. CESM-LE differs from reanalyses by a much longer time-series and the aspect that it is not constrained by observations, allowing projections. The purpose of the study is to understand which Arctic factors influence the Baltic Sea, how strong these connections are, how is AA affecting these relationships, and if the knowledge can be used in long term weather forecasting for next some months in the Baltic Sea region.

The paper is organised as follows: Section 2 describes the used datasets and methodology. Section 3 explains the results of the spatial correlations of climatic variables (stationary, 20-year periods up to the year 2100 and lagged correlations), whereas Section 4 provides a discussion of the results and conclusions.

## 2 Data and methodology

We used the CESM Large Ensemble Project (CESM-LE) set of climate model simulations on a 1°×1° horizontal grid (Kay et al., 2015). The CESM1 is a fully coupled model described by Hurrell et al. (2013). A control simulation under pre-industrial

(1850) radiative forcing conditions was run for 1800 years. A single ensemble member was branched from this control and ran from 1850 to 1920 with transient forcing. A 40-member ensemble was then performed for the period 1920-2100. All 40

CESM-LE ensemble members use the same model and the same external forcing. Each ensemble member has a unique climate trajectory because of small round-off level differences in their initial atmospheric conditions. Simply put, the CESM-LE ensemble spread results from internally generated climate variability alone (Kay et al., 2015). Each member is subject to the same radiative forcing scenario (historical up to 2005 and RCP8.5 thereafter).

We used Pearson's correlation coefficient to measure the dependence between two variables. To measure the strength of a

relationship between two variables without the possible controlling effect of a third variable, we used partial correlation:

$$R(X,Y|Z) = \frac{R(X,Y) - R(X,Z) \cdot R(Y,Z)}{\sqrt{(1 - R^2(X,Z)) \cdot (1 - R^2(Y,Z))}}, \tag{1}$$

where R(A, B) is the regular Pearson correlation. Partial correlation difference from Pearson correlation reveals the controlling factor Z effect on input variables X and Y.

The following parameters were analysed: the temperature at 2-meter level (T2m); sea-level pressure (SLP); sea ice

concentration (SIC); precipitations (PREC) as the sum of large-scale and convective precipitation; geopotential height at 500 hPa (Z500); and wind speed at 10-meter level (U10). The following large-scale indices of the atmospheric circulation – North Atlantic Oscillation (NAO), Arctic Oscillation (AO) and Barents Oscillation (BO) were calculated from the model data using eofs.xarray module in Python. NAO is defined as EOF–1 of seasonal SLP anomalies for 20–80N, 80W–40E, BO as EOF–2 of seasonal SLP anomalies for 30–90N, 90W–90E, and AO as EOF–1 of seasonal geopotential anomalies for 20–90N.

Correlations with and without the effect of the large-scale indices of the atmospheric circulation were analysed. For that purpose, partial correlations between atmospheric variables with the controlling impact of the teleconnection index were calculated.

For the testing area (TA), we chose a region around the Baltic Sea (50–65N, 10–40E). For the Arctic, we looked at the area north of 60N. We chose seven important areas (IA) in the Arctic, where the correlations with TA were stronger, to analyse the

relationships between regions in the Arctic and TA (Fig. 1, Table 1). Initially, we also looked at the Chukchi Sea and Canada Basin regions, but the correlations between climate parameters between TA and these regions were clearly weaker.

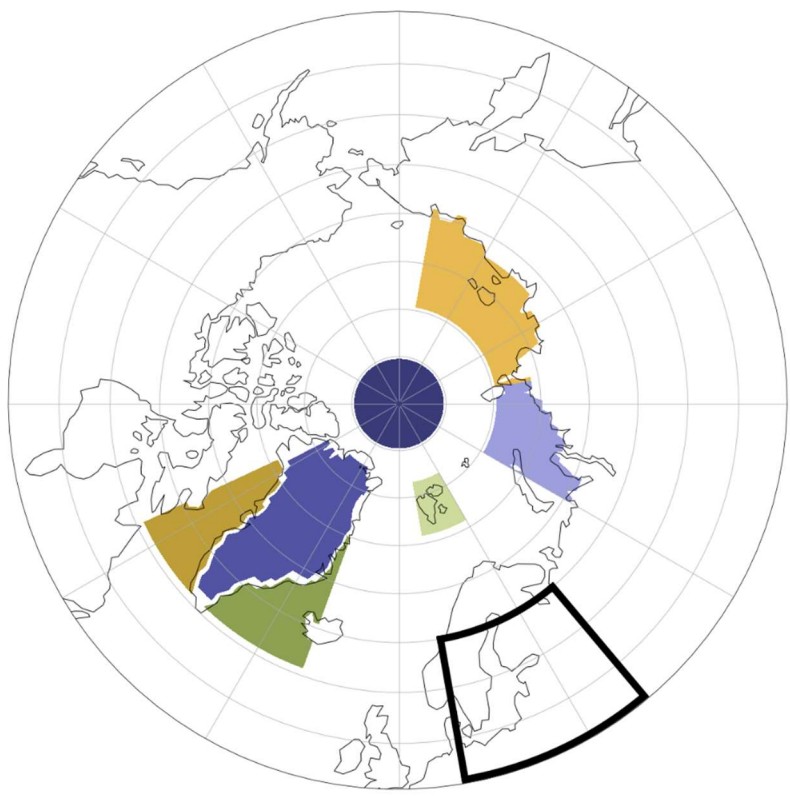

**Figure 1: Important areas (IA, coloured areas) in the Arctic and test area (TA, black rectangle).**

**Table 1. Important areas (IA)**

|   | Area | Lat 1 | Lat 2 | Lon1 | Lon 2 | Mask |
|---|------|-------|-------|------|-------|------|
| 1 | Central Arctic | 85 | 90 | 0 | 360 | No |
| 2 | Greenland | 60 | 85 | 297 | 340 | Land |
| 3 | West-Greenland | 60 | 77 | 295 | 315 | Sea |
| 4 | East-Greenland | 60 | 75 | 315 | 340 | Sea |
| 5 | Svalbard | 75 | 82 | 10 | 30 | No |
| 6 | Kara Sea | 66 | 80 | 60 | 100 | Sea |
| 7 | Laptev Sea | 66 | 80 | 100 | 170 | Sea |

We assessed the control simulation to reveal statistically significant seasonal correlations between TA and the Arctic region. For control simulation 1800-year long time-series, all correlations stronger than ±0.046 are statistically significant at the confidence level of 95 %. To analyse the strength and shape changes in teleconnections during climate change, we looked at 20-year periods of the ensemble simulation from 1980 to 2100. For every ensemble simulation, 20-year long periods with 40 ensemble members (in a total of 800 events), all correlations stronger than ±0.069 are statistically significant at the confidence level of 95 %. In order to make the connections clearer, we do not show weaker correlations than ±0.1, though they were still statistically significant. For control simulation, this gives a confidence level of 99.998 %, while for ensemble simulations of 20-year-long series, the confidence level is 99.5 %. Still, the relationship shown by the correlations may not be causal; there can be indirect connections, and we cannot rule out model internal feedback-generated correlations.

For understanding correlation maps, the first correlation parameter marks always the TA areal average, and the second one is from the Arctic. We also calculated correlations between TA and IA areal seasonal averages. We computed correlations with IA's previous month's averages to investigate the Arctic region's capability to forecast TA seasonal averages.

Ridge regression (Saleh et al., 2019) is a multiple-regression method developed for cases with a strong correlation between input parameters. We used the Ridge regression method to estimate IA-s collective forecasting capability, using the module sklearn.linear_model in Python with complexity parameter alpha = 1.0.

## 3 Results

### 3.1 Stationary spatial correlations of climatic variables

The climatic variables of separate areas are usually dependent, but the strength of the correlation depends on the distance and concrete variables. In addition to natural spatial correlations between climatic variables within near distances, also correlations emerge within longer distances. To reveal stationary connections between remote regions, we used CESM-LE 1800 year-long stationary control run. This model run is very stable without significant trends, as the radiative forcing conditions were constantly at the year 1850 level during the whole period.

We found significant correlations between several variables between the Baltic Sea testing area (TA; shown with the rectangle in our Figures) and different Arctic areas. The most robust results were found for T2m: the negative correlation between TA and the Greenland area is R < –0.6 in winter and only slightly weaker in spring and autumn (Fig. 2, row 1).

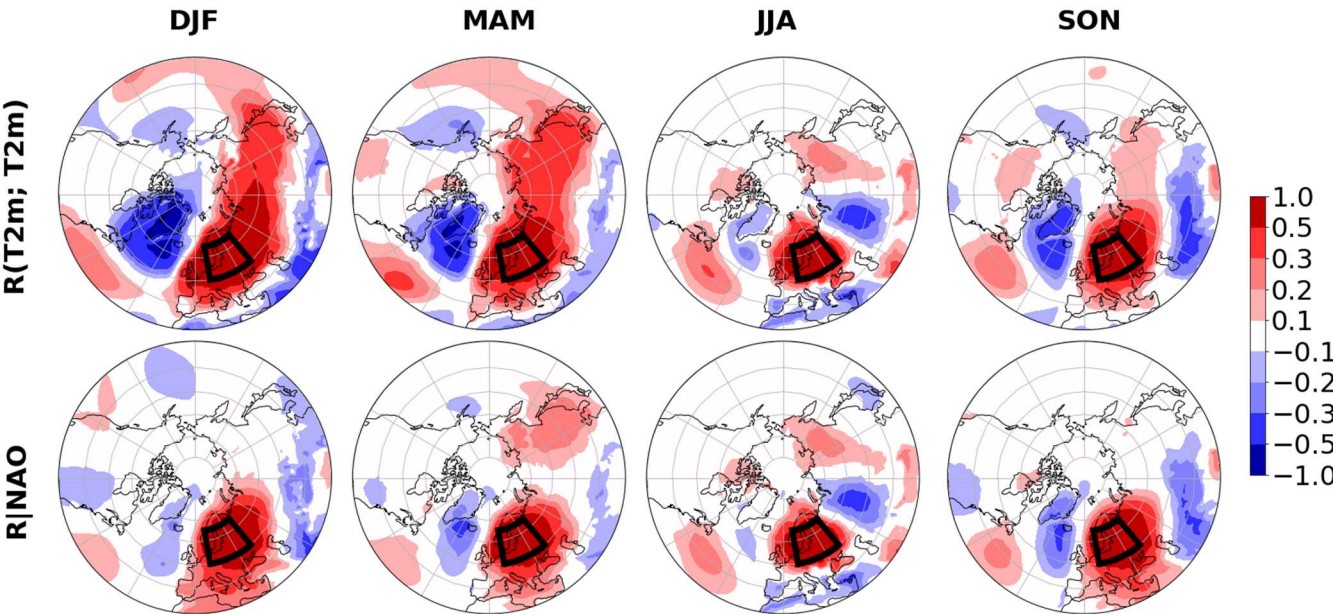

**Figure 2: The 2-meter temperature (T2m) seasonal correlations between the testing area (black rectangle) and surrounding areas according to the CESM-LE 1800-year control run. The second row shows the partial correlation with the controlling factor NAO.**

The summer is more influenced by local circulations, and remote correlations are weak. Most of these correlations are related to the NAO index – the same partial correlation with the controlling factor NAO has values in the Greenland area in winter weaker than –0.2 (Fig. 2, row 2) and in spring and autumn mostly weaker than –0.3. Teleconnections with TA that are not controlled by the NAO index are strongest at the Atlantic Ocean east to Iceland, but the correlation strength exceeds 0.3 only

in spring and autumn. The AO index effect is similar to the NAO index, with a slightly weaker impact in the Iceland region but stronger in Siberia (not shown). The BO index effect was clearly lower than that of NAO and AO (not shown). The volume of physics-based analysis of these correlations does not fit in this paper, some methods and ideas can be found in Luo et al. (2016). TA parameters correlations with Z500 or U10 were weaker than with T2m, SIC and PREC and were not included in the following analysis.

**Table 2. Seasonal correlations of control run 1800 years-long data between areal averages of testing area (TA, rows) and important area (IA, columns) parameters. The first column is the correlation between the parameter in TA and its estimation using Ridge regression over all parameters in IA-s. All shown correlations are statistically significant at the 95 % confidence level. For each TA parameter in each IA-s, the strongest correlation is highlighted in bold.**

| | | Correl | Central Arctic | | | Greenland | | | West-Greenland | | | East-Greenland | | | Svalbard | | | Kara Sea | | | Laptev Sea | | |
|---|---|---|---|---|---|---|---|---|---|---|---|---|---|---|---|---|---|---|---|---|---|---|---|
| | | Ridge | SLP | T2m | SIC | SLP | T2m | SIC | SLP | T2m | SIC | SLP | T2m | SIC | SLP | T2m | SIC | SLP | T2m | SIC | SLP | T2m | SIC |
| MAM | T2m (TA) | 0.68 | -0.42 | | 0.05 | -0.49 | -0.55 | 0.36 | -0.39 | -0.51 | 0.14 | -0.52 | -0.49 | 0.33 | -0.54 | 0.07 | -0.07 | -0.41 | 0.43 | -0.15 | -0.38 | **0.39** | -0.27 |
| JJA | | 0.54 | -0.31 | | -0.13 | -0.25 | -0.17 | 0.13 | -0.18 | -0.21 | 0.17 | -0.28 | -0.1 | | -0.29 | 0.23 | -0.29 | -0.15 | -0.12 | 0.06 | -0.17 | 0.19 | -0.16 |
| SON | | 0.59 | -0.21 | -0.09 | | -0.28 | -0.47 | 0.16 | -0.15 | -0.38 | 0.14 | -0.34 | -0.41 | 0.08 | -0.42 | 0.09 | -0.13 | -0.13 | 0.19 | | | 0.06 | |
| DJF | | 0.8 | **-0.47** | -0.2 | 0.29 | -0.58 | **-0.65** | 0.28 | -0.46 | **-0.61** | 0.21 | **-0.61** | -0.57 | 0.23 | **-0.68** | -0.07 | | -0.52 | 0.47 | -0.28 | -0.37 | 0.2 | -0.31 |
| MAM | SLP (TA) | 0.51 | | 0.11 | | -0.09 | 0.16 | -0.07 | -0.11 | | 0.08 | -0.07 | 0.28 | -0.13 | 0.08 | **0.31** | -0.22 | 0.21 | -0.08 | | 0.17 | | |
| JJA | | 0.47 | **-0.19** | -0.06 | -0.06 | -0.24 | -0.24 | 0.06 | -0.16 | -0.19 | | -0.27 | -0.17 | | -0.28 | 0.09 | -0.07 | | 0.05 | | 0.09 | 0.13 | -0.05 |
| SON | | 0.6 | -0.08 | | | -0.31 | 0.08 | 0.08 | -0.3 | -0.14 | 0.12 | -0.31 | 0.28 | | | 0.25 | -0.1 | 0.08 | -0.12 | | | | -0.07 |
| DJF | | 0.66 | 0.1 | 0.13 | -0.23 | 0.12 | **0.39** | -0.12 | 0.07 | **0.21** | | 0.15 | **0.47** | -0.15 | 0.28 | 0.29 | -0.17 | **0.31** | -0.19 | 0.07 | **0.18** | | |
| MAM | PREC (TA) | 0.47 | 0.12 | | | 0.28 | 0.05 | | 0.28 | 0.22 | -0.12 | 0.27 | -0.11 | 0.06 | 0.13 | -0.23 | 0.18 | | | | | -0.08 | 0.08 |
| JJA | | 0.53 | **0.22** | | | 0.35 | 0.26 | -0.12 | 0.28 | 0.27 | -0.13 | 0.37 | 0.16 | | **0.37** | -0.12 | 0.12 | **0.22** | -0.09 | | | **-0.17** | 0.07 |
| SON | | 0.56 | 0.16 | | | **0.39** | 0.02 | -0.1 | **0.37** | 0.2 | -0.15 | 0.38 | -0.16 | | 0.19 | -0.15 | 0.07 | 0.11 | 0.07 | | 0.1 | | 0.06 |
| DJF | | 0.59 | -0.06 | -0.07 | 0.17 | | -0.31 | 0.09 | | -0.14 | -0.05 | -0.07 | **-0.41** | 0.13 | -0.19 | -0.22 | 0.15 | -0.19 | 0.2 | | -0.12 | | 0.06 |

To generalise the results, we divided the Arctic region into important areas (IA, Fig. 1) and calculated correlations between IA and TA seasonal averages (Table 2).

The strongest correlations for IA-s are between T2m in TA and SLP in Svalbard, and seasonally, the correlation was strongest in winter (R = –0.68). Meanwhile, correlations with Svalbard T2m and SIC are much weaker with |R| < 0.31 in all seasons. TA T2m correlations with the Greenland region IA-s SLP are comparable to Svalbard, but correlations with T2m are much stronger. The correlations between T2m values in the Greenland region in winter and spring are –0.65 ≤ R ≤ – 0.49 and in autumn –0.47 ≤ R ≤ –0.38.

Correlations between TA and IA-s parameters are the weakest in summer, where the strongest correlations are R = 0.37 between PREC in TA and SLP in both East Greenland and Svalbard. Correlations in summer between T2m in TA and SLP are even weaker, with the strongest correlation R = –0.31 in the Central Arctic. SIC in IA-s has generally weaker correlations with TA parameters than SLP or T2m.

Central Arctic parameters correlation with TA is stronger than 0.4 only between TA T2m and Central Arctic SLP in winter and spring. Laptev Sea parameters correlations with TA parameters were weaker than in other IA-s, with a correlation stronger than |R| > 0.3 occurring with TA T2m only in spring and winter.

To estimate IA-s collective forecasting capability, we calculated the correlation between the parameter in TA and its Ridge regression estimation (Table 2, first column). All IA-s seasonal SLP, T2m and SIC values were used for the Ridge regression estimation. The correlation between T2m in TA and its Ridge regression estimation varies from 0.54 in summer to 0.8 in winter. The correlation between SLP in TA and its Ridge regression estimation is from 0.47 in summer to 0.66 in winter. The correlation between PREC in TA and its Ridge regression estimation is from 0.47 in spring to 0.59 in winter. As correlation

square is the measure of the proportion of variance explained, then 0.82 = 64 % of T2m variability in TA in winter can be explained by SLP, T2m and SIC variability in Arctic IA-s. The weakest connection with the Arctic is TA SLP in summer and PREC in spring, with $R^2 = 0.472 = 22$ %.

Local correlations (at the same spot, no figures shown) are necessary to better understand teleconnections between different parameters. The abovementioned strong correlation between T2m in the Greenland region and TA can be connected through local relationships with other parameters. There is a robust local connection between T2m and SIC, especially in the areas of ice margin (R~–0.9). The SIC correlation between TA and the Greenland region reaches R = –0.4; the correlation between T2m at TA and SIC in the Greenland region reaches R = 0.5.

The local correlation between SIC and U10 is mostly strongly negative (strength up to –0.8, not shown), especially in the regions where SIC is lower than 0.8. No significant local correlation exists between SIC and SLP in the Greenland Sea.

## 3.2 Spatial correlations of climatic variables during 2020–2100

To analyse how teleconnections are modified by climate change, we investigated the differences between 20-year periods and the control run. Depending on the variable, the correlation might change its spatial pattern and value between different 20-year

periods. The correlations between the following variables were analysed: T2m, SLP, SIC, PREC, and Z500. To get statistically significant results, we used for every 20-year period all 40 ensemble members, so we had a total of 800 values for each period. Most of the correlations did not show significant changes in 20-year periods from the control run, including the most emphasised correlation between T2m in TA and the Arctic region.

However, there are some statistically significant changes in correlations between T2m in the TA and SIC in the Arctic. Positive

correlations between T2m in the TA and SIC in winter (DJF) show simultaneously a remarkable weakening in the North Pole region but a significant strengthening in the Davis Strait and Hudson Bay region (Fig. 3).

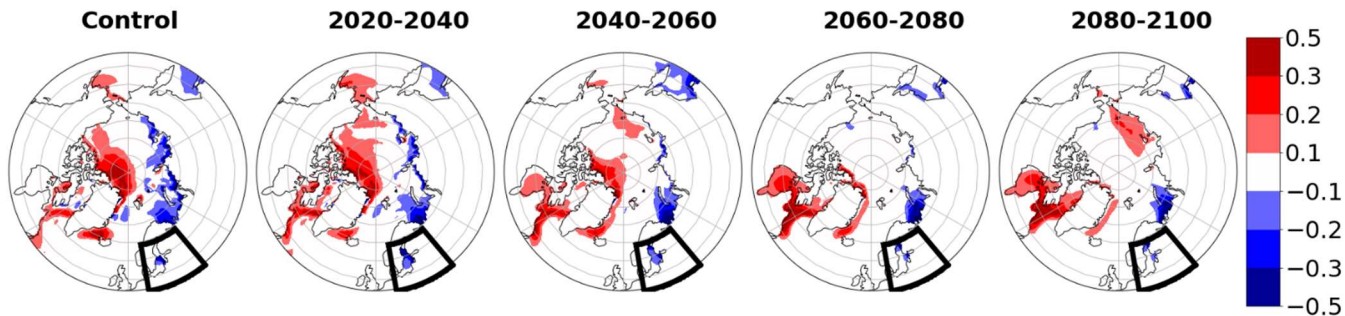

**Figure 3: The correlation in DJF between 2-meter temperature (T2m) in the testing area (black box) and sea ice concentration (SIC)**
**in the control run and 20-year periods from ensemble simulations at the period 2020–2100.**

The strong positive correlation in the Davis Strait is also remarkable in spring (MAM), but it does not strengthen as much as in winter (not shown). The correlation in winter in the region between Greenland and Iceland weakens. The negative correlation between T2m and SIC in the control run in the coastal areas of Russia becomes negligible after 2040, except in the Barents Sea, where it strengthens. A positive correlation is found in the East Siberian Sea in 2080–2100 (Fig. 3), supposedly connected with the decrease in the SIC (Fig. 4).

In regions of changing correlations, the average SIC in winter will be lessening in the Barents Sea, Hudson Bay and between Greenland and Iceland (Fig. 4). SIC in the North Pole region and coastal areas of Siberia eastward of the Barents Sea do not decrease in winter.

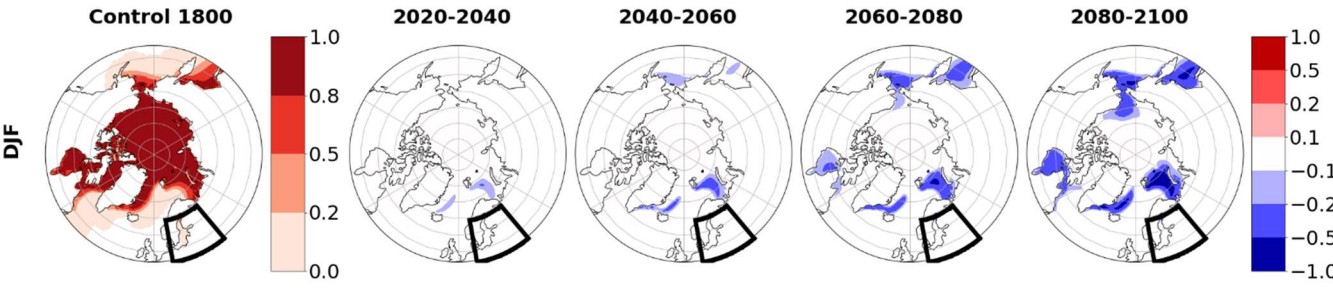

**Figure 4: The difference of sea ice concentration (SIC) 20-year average in DJF from the control run average (first column) at the 20-year long periods of 2020–2100.**

### 3.3 Lagged correlations

We are interested in factors driving variations in the Baltic Sea region and whether prior conditions may provide predictive capability. Given this, we studied the connections between earlier months (November for DJF, etc) average values of different parameters in the Arctic region and seasonal values of TA conditions. The strongest correlations were between spring (MAM) T2m in TA and the previous month's (February) average T2m and SLP (Fig. 5). The correlation values were –0.22 in Svalbard and Greenland regions; correlations between other parameters showed a weaker correlation (|R|<0.2). Analogous correlations during other seasons were weaker. We used Ridge regression to determine the predictive capability of all previous months'

average SLP, T2m and SIC in all IA-s to TA next seasonal condition. Using Ridge regression did not improve the predictability much – the strongest correlation was R = 0.30 for T2m in MAM. Thus, even for the best case, the previous month's average

values over the Arctic describe less than 10 % of the variance of next season's TA average climate state variance.

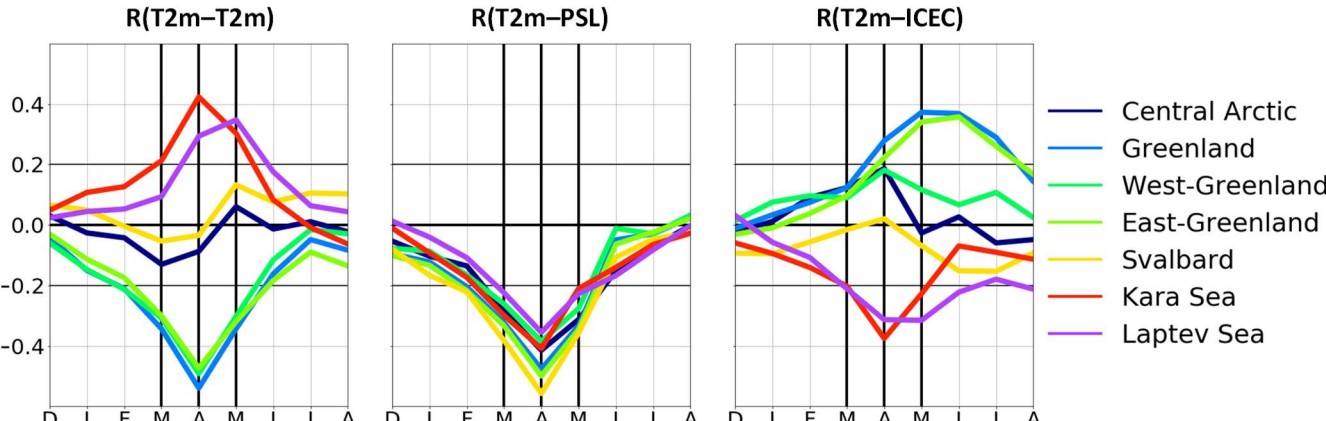

**Figure 5: Lagged correlation between spring (MAM) mean 2-meter temperature (T2m) in TA and IA-s monthly means of 1) T2m (left); 2) sea level pressure (SLP, middle); and 3) sea ice concentration (SIC, right). On the x-axis are the monthly means in IA-s. All**
**correlations stronger than ±0.046 are statistically significant at the confidence level of 95 %.**

We focused on the testing area and searched for information at which rate the Arctic region climate parameters are statistically connected with parameters in the testing area. For several variables, the connection also worked contrariwise, where the values of the testing area variables can give information about the value of the Arctic region variables. For example, the spring average

T2m in TA has R > 0.35 with SIC in June at Greenland and East-Greenland. Stronger lagged correlations from TA to the Arctic can be explained by different averaging intervals – the monthly average in the Arctic has a lower influence on the next seasonal average in the TA than the TA seasonal average to the following month's average in the Arctic.

## 4 Conclusion and discussion

The advantage of this study is the length of the stationary 1800-year long CESM-LE control database, which also reveals
relations with a weaker strength (correlations stronger than ±0.046 are statistically significant at the confidence level of 95 %). CESM-LE 40-member ensemble forecast until 2100 allows us to investigate how relationships may change in the changing climate. The most important teleconnections for the testing area are T2m, SLP and SIC in regions around Greenland and Svalbard (Fig. 1, Table 1). Our results confirmed that the Greenland region is the most influential of all Arctic regions on teleconnection to the Baltic region (as also found by Zhuo et al., 2023) but didn't confirm the old hypothesis that the mean
winter conditions over Europe depend on the summer sea ice extent in the Greenland (Hildebrandsson 1914). The lagged

correlation between summer SIC in the Greenland Sea did not significantly correlate with any primary TA parameter in the following winter. It has to be considered that our testing area is only part of the area Hildebrandsson investigated.

As far as we know, the first attempt to reveal the teleconnections between the Baltic Sea region and the Arctic was made by our workgroup in 2017 (Jakobson et al., 2017). It was based on NCEP-CFSR, and ERA-Interim reanalyses models for 1979–2015. Differences between the model parameters and different periods from the CESM-LE ensemble complicated the comparison with the present study. The comparison of T2m in the present study and temperature at the 1000 hPa level in the study made in 2017 showed a different extent in the Greenland region but similar negative correlation strength in winter. In the present study, spring and autumn showed a much stronger correlation in much wider regions around Greenland. Summer showed a very weak correlation in both studies, probably due to the more emphasised local circulation.

In investigating the influence of climatic parameters of the Arctic region on the testing area, we have to consider also the local correlations. Our results from the 1800-year long CESM-LE ensemble confirmed a strong (R∼–0.9) local connection between T2m and SIC, as found in many other studies (e.g. Olonscheck et al., 2019; Vihma et al., 2014; Outten and Esau, 2012). Our results also confirm the strong negative local correlation between SIC and U10 in the ice margin regions (strength up to –0.8), as shown in Jakobson et al. (2019). The negative correlation between SIC and W10 originates from the reduction of both stratification and aerodynamic surface roughness with a reduction of SIC (Vavrus and Alkama, 2022; Jakobson et al., 2019). Many scientists have found a lower SLP over the shrunk ice areas (Cassano et al., 2013; Alexander et al., 2004; Deser et al., 2000; Agnew, 1993), suggesting increased surface heating as a possible cause. Although Deser et al. (2000) found that mean SLP has decreased over the retracted ice margin in the Greenland Sea (according to 1958–97 reanalysis products), our results did not show a significant correlation between SIC and SLP in the Greenland Sea region during any season. According to Agnew (1993), the reason why the correlation is not present in the Greenland Sea may be due to the important role that ice export through Fram Strait and ocean currents play in determining ice extent in this region.

Barents Oscillation (BO) is related to natural variability (the variation that humans do not cause) of Arctic surface air temperature (SAT) through meridional flow and zonal wind anomalies (Chen et al., 2013). Our T2m–T2m correlation pattern in winter (Fig. 2) was similar to the BO winter pattern. We tested the BO index influence on correlations between Arctic and TA, using partial correlation. The BO index effect was insignificant or more negligible than the NAO and AO index influence on all parameters we checked in all seasons. NAO and AO index had the largest impact in winter. In summer, the local effects are more dominant, and climate indices influence is weaker.

We also aimed to reveal the ongoing climate change, especially AA's influence on teleconnections between TA and the Arctic until 2100. Most of the correlations of 20-year periods did not show remarkable differences from the control run. Changes in the TA T2m correlations with SIC in the Arctic concur with negative trends in SIC and positive trends in T2m in TA. Changes in the correlations in winter with regions with high and stable SIC values are hard to suspect to have any direct physical basis. The strongest correlation between TA and Arctic region parameters was T2m in TA and T2m in the Greenland region (Table 2). This correlation is constant up to 2100 (not shown). Sun et al. (2016) declared that the "Warm Arctic, Cold Continents" regime is transient and becoming increasingly unlikely as the climate continues to warm. There seems to be a discrepancy

between these two results, but it is not necessary. A strong negative correlation in winter means that warmer than average Arctic concur with colder than average Baltic Sea region; it does not exclude that both regions' climate can continue to warm simultaneously.

To generalise separate Arctic regions' statistical connections with the TA, we used The Ridge regression. SLP, T2m, and SIC variability in Arctic IA-s can explain from 22 % of spring PREC to 64 % of winter T2m variability in TA. Climate indices can explain a substantial amount of it. The previous month's IA-s averages forecasting capacity for TA seasonal average is much weaker – TA spring T2m has the highest coefficient of determination $R^2 = 9$ % with the Ridge regression estimation. Thus, we have to conclude that using Arctic climate data could not improve the Baltic Sea region's weather forecasting.

In conclusion – the Baltic Sea region climate has strong teleconnections with the Arctic climate; the strongest connections are with the Svalbard and Greenland region. There is high seasonality in the teleconnections, with the strongest correlations in winter and the weakest correlations in summer, when the local meteorological factors are stronger. The majority of teleconnections in winter and spring can be explained by climate indexes NAO and AO.

By the end of the 21st century, the Arctic ice concentration will significantly decrease. There will also be slight changes in the teleconnection locations and strength. Still, the general teleconnections pattern between the Arctic and the Baltic Sea region will not change during the 21st-century climate change.

The most important Arctic factors influencing the Baltic Sea are T2m and SLP, and the most important Arctic regions are Greenland and Svalbard, but the mechanisms for these teleconnections remain unknown. We have to agree with Overland et al. (2016) that there are no simple cause-and-effect pathways in the Arctic and mid-latitude weather and climate teleconnections.

**Data availability**

The CESM1-LE data are available online at https://www.cesm.ucar.edu/community-projects/lens/data-sets

**Author contributions**

EJ and LJ designed the experiments, analysed the results and wrote the paper. EJ performed the experiments and generated figures.

**Competing interests**

The contact author has declared that none of the authors has any competing interests.

**Acknowledgements**

We thank Marika Holland, Frederic Castruccio and the whole NCAR Oceanography Section for hosting our visit, for access and support to use NCAR supercomputers and for the helpful discussions.

**Financial support**

We thank Baltic-American Freedom Foundation for financing EJ visit to NCAR via BAFF Research Scholar Program.

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
