# Peer review of "Atmospheric teleconnections between the Arctic and the Baltic Sea region as simulated by CESM1-LE"

_EGUsphere, 2023_

## Author Comment (AC1)

**Referee 1**

Our responses to your comments are marked in *italic* below.

The manuscript "Atmospheric teleconnections between the Arctic and the Baltic Sea region as simulated by CESM1-LE" by Jakobson and Jakobson examines North Atlantic sea ice teleconnections to the Baltic Sea through the large-scale atmospheric pathways described by the Arctic Oscillation, North Atlantic Oscillation, and Barents Oscillation. The authors examine stationary (pre-industrial) versus modern climate forcing (RCP8.5 scenario) in their model simulations in an attempt to understanding natural versus anthropogenic forcing on these Arctic-Baltic physical connections.

Key findings are that Svalbard and Greenland Sea regions two-meter air temperature and surface pressure exhibit the strongest correlative relationships with the Baltic region climate, namely in winter due to stronger forcing by the modes of variability (NAO and AO) relative to the summer season. Under continued climate change from greenhouse gas emissions, the authors note that the end-of-century projections suggest these Arctic-Baltic relationships will remain rather consistent through time despite continued Arctic warming and sea ice loss.

The topic of Arctic change linkages with northern European climate is one that continues to receive much attention. That said, the authors could do a better job of **reviewing this work to date**, and bringing attention to what value is **added by their new analysis and modeling approach.** The methods, especially the ridge regression approach, could be more clearly defined and the results could be much more clearly stated. Moreover, the paper is difficult to follow due to **numerous grammatical errors and redundancies in uncommon acronyms**. The paper may benefit from English editing services. Several remarks along these lines are made by manuscript line number (L) in the comments that follow.

Major comments:

L105: Are the correlative results sensitive to changes in the Baltic and Arctic marginal seas domains? Did you test for this? Why were these geographic areas selected? More details are needed to provide some context for results. Citing previous literature that has used these or similar domains may help in this regard.

*Thank you for the comment. We added to the text that these domains were selected from regions where the correlations with TA were stronger, not from previous literature. Initially, we checked that, indeed – when we selected IA at a smaller region from a high correlation area, then the correlation was higher than from the case when we increased the region so that lower correlation areas were also included.*

L155: While the correlative approaches are defined in Section 2, the Ridge Regression approach is not clearly described. What does it entail and why is it used? Readers will generally be familiar with correlation techniques, but less so with this specific regression method. It should be described in detail, including with justification for why it was selected

in lieu of a simple parametric test (i.e., linear regression) as are used in the correlation analyses.

*The Ridge Regression was selected to mitigate the problem of multicollinearity in linear regression, as supposedly there are strong correlations between different regions' parameters. We added a sentence with a citation to the Ridge Regression: "Ridge regression (Saleh et al., 2019) is a multiple-regression method developed for cases when there is strong correlation between input parameters."*

Minor comments and corrections:

L8: Suggest removing "measured and"

*The correction has been made.*

L15: What is meant by "local factors"? Please be more specific.

*We replaced it with "local meteorological factors" (likewise Iguchi et al., 2018; Chen et al., 2017 used the term), which means that the global circulation influence is smaller.*

L16: Suggest revise to "NAO and AO climate indices"

*The correction has been made.*

L48: By "permanent" do you mean "seasonal" or "ephemeral" snow cover? Please clarify.

*The article we referenced used "permanent", but you are correct; „seasonal" is more precise.*

L53: Suggest substituting "knowing" for "studying" or similar word choice

*According to another referee's suggestions, we removed this paragraph.*

L75: reanalyzes ◊ reanalyses

*We chose the British English variant.*

L76: Suggest substituting "search" for "examine" or similar word choice

*The correction has been made.*

L83: Suggest substituting "completes" for "concludes" or similar word choice

*It has been reworded.*

L98: Once these climate variables are defined, they do not need to redefined (e.g., L143, L192-193, etc) through the paper. Also, suggest using typical acronyms such as SLP for sea-level pressure and SIC for sea-ice concentration as they will make it easier for readers to follow results.

*Corrected. We initially used the same acronyms the CESM model used, but we agree that typical acronyms are easier to follow.*

L102: Are the NAO, BO, and AO definitions adopted or adapted from previous studies? If so, the studies should be cited. If not, then some explanation should be used for modifying data and domains used to define these indices.

*The definitions are given L102 – L103. „NAO is defined as EOF–1 of seasonal SLP anomalies for 20–80N, 80W–40E, BO as EOF–2 of seasonal SLP anomalies for 30–90N, 90W–90E, and AO as EOF–1 of seasonal geopotential anomalies for 20–90N."*

L107: Change "above" to "north of"

*The correction has been made.*

L126: Change "supposedly not" to "less"

*The correction has been made.*

L137-138: Please reword this sentence as the second half of it is confusing.

*We reworded the sentence: "The climatic variables of separate areas are usually dependent, but the strength of the correlation depends on the distance and concrete variable."*

L161: Add "and" before "seasonally"

*The correction has been made.*

L178: "then 0.82 = 64%" – it is very difficult to follow what is meant here. Are you referencing squared correlations initially then their explained variance? Please clarify.

*We reworded the sentence: "As correlation square is the measure of the proportion of variance explained, then ..."*

L199: "positive correlation" involving what? More specifics are needed to make results easier to follow.

*We reworded the sentence: "Positive correlations between T2m in the TA and SIC in winter (DJF) show simultaneously a remarkable weakening in the North Pole region but significant strengthening in the Davis Strait and Hudson Bay region."*

L209: "earlier average month is confusing as worded" – please revise.

*We reworded the sentence: "...we studied the connections between earlier months (November for DJF, etc) average values of different parameters ...".*

L224: What is meant by "self-consistent"

*We agree, the expression is confusing. We reworded the sentence: „The advantage of this study is the length of the stationary 1800-year-long CESM-LE control database"*

L225: r=0.046 is a pretty weak threshold for a physical relationship given that a random relationship could arise ~5% of the time. Why mention this threshold?

*A random relationship could arise ~5% of the time, which is mostly a quite good rule of thumb for measured climate data and is statistically accurate when the sample size is 78. For our case, the sample size is 1800, and a random relationship could arise only ~0.2% of the time.*

L228-231: Should some sort of teleconnection hypothesis be revisited in the introduction then touched upon here? This seems like a strange place to comment that the present study does not confirm the long-proposed linkage between sea ice around Greenland and European climate.

*We moved the paragraph to the introduction and added newer references:*

*Zhuo, W., Yao, Y., Luo, D., Simmonds, I., Huang, F. 2023. The key atmospheric drivers linking regional Arctic amplification with East Asian cold extremes, Atmospheric Research, Volume 283, 2023, 106557, ISSN 0169-8095, https://doi.org/10.1016/j.atmosres.2022.106557.*

*Deng, K., Yang, S., Ting, M., Lin, A., &Wang, Z. (2018). An intensified mode of variability modulating the summer heatwaves in eastern Europe and Northern China.Geophysical Research Letters,45,11,361–11,369. https://doi.org/10.1029/2018GL079836*

L252: This sentence is confusing, please reword.

*We reworded the sentence to „Barents Oscillation (BO) is related to natural variability (the variation that humans do not cause) of Arctic surface air temperature (SAT) through meridional flow and zonal wind anomalies (Chen et al. 2013).“*

L275: Change to "will significantly decrease"

*The correction has been made.*

L278: T2m and SLP from what Arctic region are best connected with Baltic climate?

*We added, "and the most important Arctic regions are Greenland and Svalbard".*

---

## Author Comment (AC2)

**Referee 2**

**Our responses to your comments are marked in *italic* below.**

I found this paper quite difficult to follow as I was unsure of many of the methods and therefore the results made little sense. I felt that most things were not clearly explained or explained using non-scientific language and the results were not very robust thus making it hard to understand the **relevance and novelty** of this study. I have listed both the major concerns I have with this manuscript and more detailed/minor comments below that.

Major comments

I have a few issues with the methods used:

What is the ridge regression? There is no description as to what this actually does and so is difficult to understand what this means in the results and what especially when its described to be able to explain cause of winter variability in Baltic Sea.

*Thank you for paying attention to this. We added: "Ridge regression (Saleh et al., 2019) is a multiple-regression method developed for cases when there is a strong correlation between input parameters."*

For the Arctic area that you assessed, this reaches to as far as 50N. Most studies start the Arctic from **60 or 67.5N**. I think that this is especially important here because you are doing correlations of the Arctic from 50-90N to the Baltic Sea which is from 50-65N so essentially you correlate the Baltic sea area with itself when you are doing a whole Arctic correlation and so will therefore get high correlations in this region. I think it would be best to **stick to an Arctic area that is commonly used in other studies**.

*We corrected the Arctic area to be north of 60N. In the calculations, we looked already only at areas north of 60N, so no new calculations were done; the previous definition was just because we wanted to include the TA on the map.*

Due to the issues with the methods used I also have a few concerns regarding the results presented:

The description of the results were very confusing. Line 162 you say that the correlations with Svalbard T2M and ICEC are weaker, is this in the table because this is showing the correlation between TA T2M, PSL and PREC and, the IA T2M, PSL and ICEC so I am not sure how you are showing the correlation between Svalbard T2M and ICEC.

*We wrote "correlations with T2m and ICEC", not "correlations between T2m and ICEC". The correlation in this paper is always between TA and Arctic (region); also, in the table*

*caption, it is written that "Seasonal correlations of control run 1800 years-long data between areal averages of testing area (TA, rows) and important area (IA, columns) parameters."*

Figure 3 is never referenced in the manuscript and although I assume that it refers to section 3.2 it but this is not made obvious. Within this section, from assessing figure 3, if this is the correct figure, I would say that there is only a slight weakening in the winter North Pole ICEC and TA T2m correlation.

*Thank you for paying attention to this. Indeed – we had lost all citations to Figure 3 during the article writing process. The weakening at the North Pole is huge – from a range of 0.3-0.5 to less than 0.1 from 2060.*

Why did you corelate seasons to previous months particularly when in the lines 220-222 you state that a monthly correlation will have a weaker influence on the next seasonal average, why not then do solely seasonal lagged correlations or monthly lagged correlations?

*Initially, we tried doing seasonal lagged correlations, but these were even weaker than the monthly ones. We didn't do monthly lagged correlations as we were interested in seasonal averages, not monthly ones.*

Finally you mention that you analyse Z500 and U10 and yet this is not shown anywhere in the manuscript. I advise either add figures to show this or remove the reference to analysing this completely.

*We added to the result a conclusion that "TA parameters correlations with Z500 or U10 were weaker than with T2m, SIC and PREC and were not included in the following analysis."*

**Minor/detailed comments**

Line 17: Unsure of what you mean about how teleconnection patterns did not show remarkable developments with ongoing climate change, particularly the phrase 'remarkable developments'

*We reworded the sentence: "During ongoing climate change, the teleconnection patterns did not show remarkable changes by the end of the 21st century."*

Line 36: There have been many studies that have discussed Arctic amplification and the causes behind it that should be referenced here.

*We added a paragraph: The faster warming in the Arctic compared to the global mean, a phenomenon known as Arctic amplification (AA), is a result of interacting processes: sea ice loss and surface albedo feedback (Laîné et al., 2016; Yoshimori et al., 2014; Serreze et al., 2009; Screen and Simmonds 2010), changes in longwave and/or temperature feedbacks (Dai and Jenkins 2023; Jenkins and Dai 2021; Duan et al., 2019; Pithan and Mauritzen 2014; Lu and Cai 2009), cloud changes (Taylor et al., 2022; Boeke and Taylor 2018; Taylor et al., 2015; Taylor et al., 2013; Francis and Hunter 2006; Vavrus 2004), intraseasonal cycling of heat (Clark et al., 2021; Bintanja and Krikken 2016; Bintanja and Linden 2013), and*

*poleward energy transport (Sang et al., 2022; Spielhagen et al., 2011). However, the relative weight of these different factors is still under debate (Taylor et al., 2022; Dai et al., 2019).*

Line 39: What is meant by; 'AA is expected to be related to further changes that affect mid-latitudes'

*The idea is that AA will not only stay in the Arctic but will also influence mid-latitude climate change.*

Line 54: 'Possibility to glance in to the future' this is not accurate (or scientific enough) as to what scientists do when assessing the impact of Arctic on mid-latitudes. We assess the impact of Arctic on mid-latitudes to improve accuracy for forecasting or climate projections, we cannot glance in to the future but give a scenario based prediction of what may occur.

*We removed this paragraph.*

Lines 54- 62: I do not understand the relevance of this section. I suggest either removing it or being explicit about why the reader needs to know about the pace of climate change.

*We removed this paragraph.*

Line 66: This sounds like you have multiple models but with the CESM-LE you have one model that has multiple members.

*There are several large ensembles with individual models and yes – we use only one of them.*

Line 101: Can you give references to the classification of the AO, NAO and in particular the Barents Oscillation as it is less well known that the other two.

*The definitions are given L102 – L103. „NAO is defined as EOF–1 of seasonal SLP anomalies for 20–80N, 80W–40E, BO as EOF–2 of seasonal SLP anomalies for 30–90N, 90W–90E, and AO as EOF–1 of seasonal geopotential anomalies for 20–90N. "*

Line 104: Can you be more explicit here in your methods as to how the 'correlations with and without the effect of teleconnection indices' were analysed

*We used partial correlation, defined on lines 94-97. For the controlling third variable, we used teleconnection indices.*

Line 109: 'were clearly weaker than with the remaining regions' feels like there is a word(s) missing here.

*We reworded the sentence: "Initially, we also looked at the Chukchi Sea and Canada Basin regions, but the correlations between climate parameters between TA and these regions were clearly weaker".*

Line 123: What do you mean by teleconnection transformations?

*We reworded the sentence: " To analyse the strength and shape changes in teleconnections during climate change, we looked at 20-year periods of the ensemble simulation from 1980 to 2100."*

Line 126: 'weaker correlations are supposedly not important' – yes stronger correlations are more important

*We reworded the sentence: "In order to make the connections clearer, we do not show weaker correlations than ±0.1, though they were still statistically significant."*

Line 137: This line is very confusing, please edit.

*We reworded the sentence: "The climatic variables of separate areas are usually dependent, but the strength of the correlation depends on the distance and concrete variable".*

Line 147: Is row 1 or 2 the one that is not controlled by NAO and what is the sign of the strong teleconnection in the Atlantic to the east of Iceland, in my view the strongest correlations are off the coast of North America.

*The first row is the regular correlation. The second row is the partial correlation with the controlling factor NAO, which means that the second row would be the correlation when there would be no variability in NAO. The strong teleconnection in the Atlantic at the region around 60N, 30W is negative.*

Table 2: Why did you separate East and West Greenland

*We separated them because there are clear differences in some parameters, e.g. SLP and PREC in winter.*

Lines 181-185: I am not sure which figure you are referencing here, can you add this.

*These figures were not added. We clarified this by "Local correlations (at the same spot, no figures shown) are necessary ...".*

Line 202: The correlation becomes negligible off the coast of Siberia rather than fades away and Barents sea is not off the coast of Siberia but Russia. Also a positive correlation is found in the East Siberian Sea in 2080-2100 showing that not all correlation fades away.

*New text here: "The negative correlation between T2m and SIC in the control run in the coastal areas of Russia becomes negligible after 2040, except in the Barents Sea, where it strengthens. A positive correlation is found in the East Siberian Sea in 2080-2100 (Figure 3)".*

Line 217-218: What rate are you talking about here and what is meant by incoming parameters?

*Here, we use the rate as the percentage of the total variability that a parameter from TA can describe. The Word "incoming" is indeed accidental; we removed it from the article.*

Line 218-219: Revise this line starting 'It turned out that quite common…' as I am not sure what is meant in particular what is quite common?

*We reworded the sentence: "There were several variables the connection worked also contrariwise, where the values …"*

Line 224: What is meant by CESM-LE self-consistent database?

*We agree the expression is confusing.* We reworded the sentence: „*The advantage of this study is the length of the stationary 1800-year-long CESM-LE control database*"

Line 225: These correlations are very weak, plus this value differs from the one stated in the methods, are these values supposed to be different?

*The idea in presenting these values is to show that for such long timeseries, even quite weak correlations are statistically significant. Here, the stated value is for a database with 1800 elements. In the methods, there was also the value for 800 element database added.*

Line 230: Are there no more up to date papers than Hildebrandsson which is 109 years old?

*We moved the paragraph to the introduction and added newer references:*

*Zhuo, W., Yao, Y., Luo, D., Simmonds, I., Huang, F. 2023. The key atmospheric drivers linking regional Arctic amplification with East Asian cold extremes, Atmospheric Research, Volume 283, 2023, 106557, ISSN 0169-8095, https://doi.org/10.1016/j.atmosres.2022.106557.*

*Deng, K., Yang, S., Ting, M., Lin, A., &Wang, Z. (2018). An intensified mode of variability modulating the summer heatwaves in eastern Europe and Northern China.Geophysical Research Letters,45,11,361–11,369. https://doi.org/10.1029/2018GL079836*

Line 236: What differences in the model parameters and different periods from CESM-LE complicated a comparison with the Jacokson et al. 2017 paper? A lot of studies compare reanalysis and historical CESM simulations.

*It is quite common to do a comparison between reanalysis and climate model data. If you want to repeat analyses compiled for reanalysis with climate model data, then your choices are limited with the previous work. In our 2017 paper, we did not analyse 2 m temperature but 1000 hPa level T2m and its vertical profile, but in this paper, we were mostly interested in 2 m temperature.*

Line 237: The comparable T2m – comparable to what?

*Changed to "The comparison of T2m in the present study …"*

Line 253: What is natural SAT variability?

*We added an explanation to the natural variability: "… related to natural variability (the variation that humans do not cause) of …"*

Line 260. This sentence is confusing, please reword.

*Changed to "Changes in the TA T2m correlations with ICEC in the Arctic concur with negative trends in ICEC and positive trends in T2m in TA."*

Line 262: I am not sure what you mean here, can you reword.

*Changed to "The strongest correlation between TA and Arctic region parameters was T2m in TA and T2m in the Greenland region."*

Line 273: what local factors are you talking about?

*We replaced it with "local meteorological factors" (likewise Iguchi et al., 2018; Chen et al., 2017 used the term), which means that the global circulation influence is smaller.*

---

## Author Comment (AC3)

**Referee 3**

**Our responses to your comments are marked in *italic* below.**

This submission follows up some earlier work by these authors which made use of reanalysis data (namely NCEP and ERA-I) for the period 1979-2015 to explore teleconnections between the Arctic and the Baltic. This present work attacks a similar issue using model data. The submission has the potential to make a significant contribution to the literature, but it is not quite there yet. Before I would be able to recommend acceptance, there are a number of issues which need to be addressed.

Lines 29-30: Also here to cite the recent analysis of Mika Rantanen, Alexey Yu Karpechko, Antti Lipponen, Kalle Nordling, Otto Hyvärinen, Kimmo Ruosteenoja, Timo Vihma and Ari Laaksonen, 2022: The Arctic has warmed nearly four times faster than the globe since 1979. *Communications Earth & Environment*, **3**, 168, doi: 10.1038/s43247-022-00498-3.

*We reworded the sentence: "The Arctic region is warming at least twice (IPCC, 2021; Nakamura & Sato 2022; Overland et al., 2018; Meleshko et al., 2020), some authors showed that nearly four times (Rantanen et al., 2022) as fast as the whole planet …"*

Line 30: What 'average' is referred to here?

*Corrected: average → global average*

Line 38: 'Screens' should be 'Screen'

*The correction has been made.*

Line 42: Question raised here is to which specific region of the Arctic is of importance. Add to this reference to Wenqin Zhuo & co-authors, 2023: The key atmospheric drivers linking regional Arctic amplification with East Asian cold extremes. Atmospheric Research, 283, 106557, doi: 10.1016/j.atmosres.2022.106557.

*Thank you for the excellent reference; we added it.*

Lines 44-45: Valuable here to cite the more recent analyses of

Overland, J. E., Ballinger, T. J., Cohen, J., Francis, J. A., Hanna, E., Jaiser, R., Kim, B.-M., Kim, S.-J., Ukita, J., Vihma, T., Wang, M. and Zhang, X. 2021. 'How do intermittency and simultaneous processes obfuscate the Arctic influence on midlatitude winter extreme weather events?', *Env. Res. Lett.* **16**, 043002, doi: 10.1088/1748-9326/abdb5d,

Luo, D., X. Chen, J. Overland, et al., 2019: Weakened potential vorticity barrier linked to recent winter Arctic sea ice loss and midlatitude cold extremes. J. Climate, 32, 4235-4261, doi: 10.1175/JCLI-D-18-0449.1, and

Rudeva, & coauthors, 2021: Midlatitude winter extreme temperature events and connections with anomalies in the Arctic and tropics. J. Climate, 34, 3733-3749, doi: 10.1175/JCLI-D-20-0371.1.

*Thank you for the excellent references; we added these.*

Lines 79-80, …: The paper makes frequent allusions to 'forecasting', but it is not always clear what timescale is meant. Please make these parts more specific. One could argue that anything longer than 2-3 weeks is really an 'outlook'.

*As for forecasting, we are thinking of timescales 1 – 3 months. Line 80, we changed "long-term weather forecasting" to "long-term weather forecasting for the next couple of months".*

Lines 121-129: The value of very long simulations as used here is that it is 'easier' to achieve statistical significance. However, such long integrations may not be able to reveal PHYSICAL significance. The threshold correlations here are very small and explain less than 1% of the variance. Strongly suggest the authors add some remarks to point out these issues.

*Correlations stronger than ±0.1 are, for our database, statistically very significant, but you are right – to find physical connections behind the correlation, often higher correlations are needed. We just wanted to point out that we ignore all correlations weaker than 0.1, not that we will analyze all correlations stronger than 0.1. We changed the text to clarify it from "This paper only looks at correlations stronger than ±0.1" to "This paper only presents correlations stronger than ±0.1".*

Also, the time series considered here will possess considerable autocorrelation. This, in turn, with reduce the 'effective' degrees of freedom with which the tests for significance are conducted. Was allowance made for this effect (see, e.g., Bretherton C S et al 1999 The effective number of spatial degrees of freedom of a time-varying field J. Climate 12 1990-2009).

*When working with hourly or daily time series, then the autocorrelation is definitely important. In our paper, we compare monthly and seasonal data on a yearly basis, and there is no substantial autocorrelation, so the effective degrees of freedom are not considerably affected.*

Lines 132-134: Not all readers will be familiar with Ridge regression. Helpful here to reference the recent (and accessible) monograph of Saleh, A. K. Md. Ehsanes; Arashi, Mohammad; Kibria, B. M. Golam (2019). Theory of Ridge Regression Estimation with Applications. New York: John Wiley & Sons. ISBN 978-1-118-64461-4.

*Thank you for the hint – we hadn't noticed this book before. We added the reference to the manuscript and included a sentence to describe it.*

Lines 146-150: The demonstrated link here to the NAO is interesting. A little more physics-based discussion is required on this. Make reference here to the investigation of Luo, D., Y. Xiao, …, 2016: Impact of Ural Blocking on winter Warm Arctic–Cold Eurasian anomalies. Part II: The link to the North Atlantic Oscillation. J. Climate, 29, 3949-3971, doi: 10.1175/JCLI-D-15-0612.1.

*We believe that reasonable physics-based discussion would take too much volume (and of course a lot of additional analyses), so we hope that it is OK if we just mention this direction in the text. We added: „The volume of physics-based analysis of these correlations does not fit in this paper, some methods and ideas can be found in Luo et al (2016)."*

That paper also implies a role of the Ural Mountains, and the atmospheric blocks situated over them. The Urals are only a short distance 'downstream' of the Baltic and would be expected to influence this local region. The paper would greatly benefit from some thoughts on this aspect, and of changes and variability in blocking. Beneficial in this to cite analysis of

Luo, Dehai & coauthors, 2017: Increased quasi-stationarity and persistence of winter Ural Blocking and Eurasian extreme cold events in response to Arctic warming. Part II: A theoretical explanation. J. Climate, 30, 3569–3587, doi: 10.1175/JCLI-D-16-0262.1.

*We didn't make any calculations on Ural Mountains blockings, but the theme definitely deserves attention. We added this paragraph to the Introduction:*
*„Furthermore, some direct impacts are influenced by remote processes in the Arctic. For example, possibly the Barents and Kara Seas warming associated with the sea ice loss affects the Ural blocking (Peings et al., 2023; Yao et al., 2017; Luo et al., 2017), which has been identified as precursors of sudden stratospheric warmings (Lu et al., 2021, Statnaia et al., 2020; Lee et al., 2019; Cohen & Jones, 2011; Martius et al., 2009), and extreme temperature/precipitation anomalies over Europe (Yang et al., 2022; Peings, 2019; Cattiaux et al., 2010)."*

Lines 181-188: The 'local' correlations are similar to what one would expect. It is not clear to me that they are 'necessary to better understand the teleconnections …' Please clarify.

*We are afraid that if we don't mention the local correlations at all, some readers might get stuck with them.*

Lines 193-194: See my earlier point on sample size and 'statistically reliable results'. May wish to reword this.

*We changed the text from "To get statistically reliable results" to "To get statistically significant results". We assume that for 99.5% confidence level results, we can say without hesitation that they are statistically significant.*

6. Lines 195-206: I have trouble interpreting these results, as it is not made clear what the sea ice is doing in these simulations. The integrations to 2100 are performed with RCP8.5 forcing and the reader is entitled to some (limited) information as to how the Arctic ice is changing over the period. As the changes will almost certainly be large the correlations in the last epochs of this century will essentially refer to physical associations which are different from those in the earlier part of the 21st century. The interpretation here needs much more thought. Of relevance to these considerations is the recent analysis of Yeon-Hee Kim, Seung-Ki Min, Nathan P. Gillett, Dirk Notz and Elizaveta Malinina, 2023: Observationally-constrained projections of an ice-free Arctic even under a low emission scenario. Nature Communications, 14, 3139, doi: 10.1038/s41467-023-38511-8.

*Most of these results are about winter when there is no significant change in the ice conditions (Fig 4), only between Greenland and Iceland does the decrease in the correlation concur with the decrease in the ice concentration.*

Line 216: Insert 'of the variance' after '10%'.

*We added the variance there.*

Line 228-233: An interesting point is made here in connection with the potential influence from the Greenland Sea. It would be worth mentioning in the text that Zhuo, Yao et al., 2023: The key atmospheric drivers linking regional Arctic amplification with East Asian cold extremes. Atmospheric Research, 283, 106557 found sea ice cover this Greenland region to be one of the most influential of all Arctic regions on teleconnections to, and conditions in, the Baltic.

*We completed this paragraph, added the reference you suggested, and in addition, this reference:*
*Deng, K., Yang, S., Ting, M., Lin, A., &Wang, Z. (2018). An intensified mode of variability modulating the summer heatwaves in eastern Europe and northern China.Geophysical Research Letters,45,11,361–11,369. https://doi.org/10.1029/2018GL079836*
*We added the paragraph about Arctic regions' connections to Europe to the introduction (as suggested by another referee).*

4. Lines 243-245: What are the physics behind this correlation. Is it 'on shore' flow, Ekman effect. Also here may be informative to reference the recent work of. Vavrus, S. J., and R. Alkama, 2022: Future trends of arctic surface wind speeds and their relationship with sea ice in CMIP5 climate model simulations. Climate Dyn., 59, 1833-1848, doi: 10.1007/s00382-021-06071-6.

*We added a sentence: "The negative correlation between SIC and W10 originates from the reduction of both stratification and aerodynamic surface roughness with a reduction of SIC (Vavrus and Alkama, 2022; Jakobson et al., 2019)."*

Lines 263-265: Please to note that the year of Lantao Sun's paper is '2018', not '2016'. The reason(s) for this disagreement must be canvassed. Sun t al. used the GFDL model (with RCP8.5). Related to my earlier point part of this difference may be due to how the sea ice evolves over the century. A more critical examination is required here.

*The Sun's paper we are referring to was published online 25 MAY 2016.*

*We improved the text as follows: "This correlation is constant up to 2100 (not shown). Sun et al. (2016) declared that the "Warm Arctic, Cold Continents" regime is transient and becoming increasingly unlikely as the climate continues to warm. There seems to be a discrepancy between these two results, but it is not necessary. A strong negative correlation in winter means that warmer than average Arctic concur with colder than average Baltic Sea region, it does not exclude that both regions' climate can continue to warm at the same time."*

Line 411-416: These two reference are the same, except the dates are different. I strongly suspect that the authors meant the first of these to be

Lantao Sun, Judith Perlwitz and Martin Hoerling, 2016: What caused the recent "Warm Arctic, Cold Continents" trend pattern in winter temperatures? Geophysical Research Letters, 43, 5345-5352, doi: 10.1002/2016gl069024

*Yes, you are correct. We corrected it.*

---

## Author Response (AR2)

**Our responses to your comments are marked in *italic* below**

The authors have carefully responded to the range of points and issues raised by myself and my two co-reviewers.

There are a few matters that still require attention. Once the have been addressed I will be pleased to recommend acceptance.

I think the paper would benefit form a better introduction to the study. For example, at line 35 sea ice loss itself is mentioned here but there is no corresponding reference. Given the importance of this trend in the analysis I suggest referencing here, e.g., the recent paper of Li M (2021) Trends and variability in polar sea ice, global atmospheric circulations and baroclinicity. Ann. NY Acad. Sci. 1504: 167-186 doi: 10.1111/nyas.14673.

*Accepted, reference to the paper is added.*

Lines 36-37: The role of the downward long wave is a good point here. Add the supporting reference of Lee S, Gong T et al., 2017: Revisiting the cause of the 1989-2009 Arctic surface warming using the surface energy budget: Downward infrared radiation dominates the surface fluxes. Geophys. Res. Lett. 44: 10,654–10,661 doi: 10.1002/2017GL075375.

*Accepted, reference to the paper is added.*

Line 42: The text speaks of 'linkages' here, and in just a few other places in the document. However, 'teleconnection' is used many times (including the title). It is not clear that the authors are referring to different phenomena when using these terms. I find the use of the two terms needlessly confusing, and suggest just using 'teleconnections' throughout.

*Accepted, all 'linkages' are changed to 'teleconnections'.*

Lines 105-106: '500 mb' should be '500 hPa'.

*Corrected.*

Lines 106-107: The North Atlantic Oscillation, AO and the Barents Oscillation are not really 'teleconnection indices' (even though they do have impacts on teleconnections). Better terminology would be 'large-scale indices of the atmospheric circulation' or something similar.

Compatible change in terminology is needed at lines 110 and 111.

*We changed 'teleconnection indices' to 'large-scale indices'.*